# Early- and Late-Onset Alzheimer’s Disease: Two Sides of the Same Coin?

**DOI:** 10.3390/diseases12060110

**Published:** 2024-05-22

**Authors:** César A. Valdez-Gaxiola, Frida Rosales-Leycegui, Abigail Gaxiola-Rubio, José Miguel Moreno-Ortiz, Luis E. Figuera

**Affiliations:** 1División de Genética, Centro de Investigación Biomédica de Occidente, IMSS, Guadalajara 44340, Jalisco, Mexico; cesar.valdez2320@alumnos.udg.mx (C.A.V.-G.); frida.rosales9343@alumnos.udg.mx (F.R.-L.); 2Doctorado en Genética Humana, Centro Universitario de Ciencias de la Salud, Universidad de Guadalajara, Guadalajara 44340, Jalisco, Mexico; 3Maestría en Ciencias del Comportamiento, Instituto de Neurociencias, Centro Universitario de Ciencias Biológicas y Agropecuarias, Universidad de Guadalajara, Guadalajara 44340, Jalisco, Mexico; 4Instituto de Investigación en Ciencias Biomédicas, Centro Universitario de Ciencias de la Salud, Universidad de Guadalajara, Guadalajara 44340, Jalisco, Mexico; abigail.gaxiola2327@alumnos.udg.mx; 5Facultad de Medicina, Universidad Autónoma de Guadalajara, Zapopan 45129, Jalisco, Mexico; 6Instituto de Genética Humana “Dr. Enrique Corona Rivera”, Departamento de Biología Molecular y Genómica, Centro Universitario de Ciencias de la Salud, Universidad de Guadalajara, Guadalajara 44340, Jalisco, Mexico

**Keywords:** Alzheimer’s disease, ApoE, early onset, late onset

## Abstract

Early-onset Alzheimer’s disease (EOAD), defined as Alzheimer’s disease onset before 65 years of age, has been significantly less studied than the “classic” late-onset form (LOAD), although EOAD often presents with a more aggressive disease course, caused by variants in the *APP*, *PSEN1,* and *PSEN2* genes. EOAD has significant differences from LOAD, including encompassing diverse phenotypic manifestations, increased genetic predisposition, and variations in neuropathological burden and distribution. Phenotypically, EOAD can be manifested with non-amnestic variants, sparing the hippocampi with increased tau burden. The aim of this article is to review the different genetic bases, risk factors, pathological mechanisms, and diagnostic approaches between EOAD and LOAD and to suggest steps to further our understanding. The comprehension of the monogenic form of the disease can provide valuable insights that may serve as a roadmap for understanding the common form of the disease.

## 1. Introduction

Dementia is any disorder in which a significant decline in a person’s previous level of cognition affects their ability to perform occupational, domestic, or social activities. In general, dementia should be viewed as an acquired syndrome with multiple potential causes rather than as a distinct disease on its own [1]. A traditional approach to understanding dementia is to divide it into two broad disease categories: neurodegenerative conditions (irreversible) and non-neurodegenerative conditions (reversible) [2]. Common causes of non-neurodegenerative mild cognitive impairment and dementia which can occur at any age include vitamin deficiencies such as B12 and thiamine, chronic alcohol abuse, chemotherapy-induced cognitive dysfunction, hypothyroidism, infections, traumatic brain injury, and other psychiatric conditions such as severe depression or anxiety. However, it is important to note that most cases of dementia in the elderly population are due to some degree of neurodegeneration. Common degenerative dementias in older people include Parkinson’s disease, dementia with Lewy bodies, vascular dementia, frontotemporal lobar degeneration, and, the most common of all, Alzheimer’s disease (AD) [3].

In 1907, Alois Alzheimer penned a case report describing the condition of Auguste Deter, who sought treatment at his clinic. Deter exhibited a variety of signs and symptoms, notably including severe language difficulties and behavioral disturbances, such as paranoid delusions and anxiety. By contemporary standards, this female would be classified as having an unusual variant of AD. Furthermore, Deter was diagnosed with the disease at the age of 51, indicating that she had an early-onset form of AD. After Auguste Deter’s death in 1906, Alois Alzheimer analyzed her brain and identified “neuritic” plaques and neurofibrillary tangles, which he linked to the patient’s condition. Following this groundbreaking discovery, thorough research has shed light on the complex characteristics of AD [4]. From its initial recognition as a rare disease, it has become a global public health concern. The incidence of AD has increased significantly with the aging of the population, solidifying its status as one of the most important health challenges of the 21st century [5,6].

Today, AD is considered a progressive neurodegenerative disease characterized by the gradual deterioration of cognitive capacities and memory functions. Within the spectrum of dementia pathologies, AD is a predominant subset, accounting for an estimated 60 to 70% of dementia cases. The global impact of AD is massive, with a prevalence of more than 50 million people worldwide. It has been projected that the prevalence of dementia will triple by 2050, leading to an increasing societal burden on a global scale; in addition, the reported deaths from AD increased by approximately 148% between 2000 and 2018 [7,8,9].

The age of onset (AoO) remains clinically important in defining the observable characteristics of the disease, and AD cases have traditionally been classified into two categories: early onset (EOAD) and late onset (LOAD). This classification is based on an arbitrary age threshold that aligns with the onset of clinical symptoms. On one hand, EOAD cases represent approximately 5–10% of all AD cases. Affected individuals develop symptoms before the age of 65 years with an estimated heritability of 92–100% [5]. Furthermore, they exhibit remarkable variability in their clinical and neurobiological characteristics, requiring different approaches to their management. Some researchers even propose that the dissimilarities are significant enough to establish a distinct form of the disorder [5,10,11,12].

On the other hand, the most common initial clinical presentation of LOAD is an episodic memory deficit accompanied by progressive impairment in other cognitive domains with a heritability rate of 60–80%. For every twenty people with AD, eighteen will have LOAD and two will have EOAD. Furthermore, for every twenty early-onset patients, one of them has an autosomal dominant variant [5].

## 2. Risk Factors: Early-Onset vs. Late-Onset Alzheimer’s Disease

### 2.1. Genetics Factors

The genetic basis of AD shows a significant divergence between EOAD and LOAD.

Between 35% and 60% of EOAD patients have at least one affected first-degree relative, and in 10% to 15% of those familial EOAD patients, the mode of inheritance is autosomal dominant transmission [13]. In a significant proportion of EOAD cases, mutations in specific genes are responsible for the early onset of the disease. The three most important genes are as follows. *PSEN2* (Presenilin 2): Mutations in the *PSEN2* gene are the least common but can lead to EOAD. *PSEN1* (Presenilin 1) and *APP* (Amyloid Precursor Protein): Mutations in the *PSEN1* and *APP* genes are the most common genetic causes of EOAD. These mutations affect the processing of amyloid precursor protein, resulting in increased production of beta-amyloid (Aβ) [14]. Individuals with mutations in any of these genes have a significantly higher risk of developing EOAD, often in their 30s to 50s. The familial nature of EOAD means that there is a higher likelihood of multiple affected family members, making genetic counseling and testing an important consideration [15]. Recent whole-genome and whole-exome studies have revealed a wealth of information regarding the genetic underpinnings of EOAD. These comprehensive analyses have identified over 20 loci associated with genes participating in various metabolic pathways, with each locus contributing a modest risk to the development of EOAD. Among these, notable genes include sortilin-related receptor A (SORLA), the anti-inflammatory microglial triggering receptor expressed on myeloid cells 2 (TREM2), ATP-binding cassette subfamily A member 7 (ABCA7), and others involved in endocytosis, endolysosomal transport, immunologic reactivity, and lipid metabolism, such as PLD3, PSD2, TCIRG1, RIN3, and RUFY1. Additionally, genes like GRN, MAPT, and C9ORF72 have also been implicated in the genetic architecture of EOAD [16,17].

A significant advancement in this field is the integration of these genetic variants into a polygenic risk score, which holds promise in predicting the likelihood of developing EOAD, with accuracies ranging from 72.9% to 75.5%. This approach allows for a comprehensive assessment of an individual’s genetic susceptibility to EOAD, facilitating early identification and intervention strategies for at-risk individuals [18].

LOAD: In contrast, LOAD does not have a single dominant genetic cause; the risk variants are frequently observed in the general population but impart minimal overall risk, thereby offering limited predictive value regarding the likelihood of developing LOAD. Nonetheless, they serve a crucial role in identifying enriched pathways, providing valuable insights that guide molecular cell biologists and biochemists toward a deeper understanding of cellular processes involved in AD pathogenesis. While the *APOEε4* allele of the apolipoprotein E protein (ApoE) is a well-established risk factor for LOAD, it is nevertheless not deterministic. Having one copy of the *APOEε4* allele increases the risk, and having two copies further increases the risk [19]. However, many individuals with LOAD do not carry the *APOEε4* allele, and conversely, not all carriers of *APOEε4* develop LOAD. In addition, LOAD is polygenic, with multiple genetic variants contributing to overall susceptibility. In fact, more than 600 Alzheimer’s susceptibility genes have been identified. Genome-wide association studies (GWASs) have identified several risk genes, including *CLU*, *CR1*, *PICALM*, *BIN1*, *CD2AP*, *PICALM*, *PLD3*, and *TREM2*. Together, these variants (but not all) contribute to the complex genetic architecture of LOAD [20,21].

### 2.2. Non-Genetic Factors

Environmental and lifestyle factors such as physical fitness and cognitive stimulation, among others, can play an essential role in AD, and their impact can vary between EOAD and LOAD. This can be explained by the Scaffolding Theory of Aging and Cognition (STAC), which suggests that as people age, their brain undergoes changes that impact cognitive performance. Also, compensatory brain processes can help preserve cognitive function in healthy older adults. For instance, such processes can compensate for mild white matter disease and age-related atrophy [22].

In individuals who have not developed cognitive dysfunction but have a genetic risk for Alzheimer’s disease, the compensatory response occurs at a faster rate. However, this response decreases quickly once cognitive dysfunction and hippocampal atrophy become noticeable. On the other hand, older people who have age-appropriate cognition and a lower risk of developing AD tend to use increased brain activation to maintain their functionality [23].

EOAD: While genetic factors often play a more dominant role, environmental and lifestyle factors can still contribute to the risk of the disease, especially in individuals with a genetic predisposition. In particular, head trauma has been associated with an increased risk of EOAD. This association may be more pronounced in individuals with pathogenic variants, due to the fact that traumatic brain injury may exacerbate neurodegenerative processes already initiated by genetic mutations [20,21,22,23,24].

LOAD is influenced by a wider range of environmental and lifestyle factors. Several cardiovascular risk factors have been associated with an increased risk, including hypertension. Chronic high blood pressure has been associated with an increased risk of LOAD. It may contribute to cerebrovascular changes that may exacerbate AD-related brain pathology [25]. Diabetes: Type 2 diabetes is considered a potential risk factor for LOAD. Insulin resistance and glucose dysregulation may adversely affect brain function and contribute to AD pathology [26]. Smoking is a significant risk factor for LOAD, and it is a factor that individuals can actively change. It is thought to promote oxidative stress and inflammation, processes that may exacerbate neurodegeneration [27]. Hypercholesterolemia: Elevated cholesterol levels in midlife have been associated with an increased risk of LOAD later in life. High cholesterol may contribute to the buildup of Aβ plaques in the brain [28]. Physical Activity and diet: Engaging in regular physical activity and maintaining a heart-healthy diet may help reduce the risk of LOAD. These lifestyle choices are associated with improved cardiovascular health, which, in turn, may benefit brain health [29].

### 2.3. Other Factors

It is essential to consider various contributing factors to the onset of AD. One such substantial element that has been extensively studied is non-coding RNAs. These can be categorized based on their length into short-chain ncRNAs that include circular RNAs, short-interfering RNAs, microRNAs, piwi-associated RNAs, and long ncRNA (lncRNA). Long non-coding RNA, as the name indicates, refers to species longer than 200 nucleotides in length. It is a critical player in many biological processes, including epigenetic control of chromatin modification, mRNA stability, and promoter-specific regulation of genes. MicroRNAs typically consist of 21–23 nucleotides and exert gene-regulatory functions by forming sequence-specific base pairs with mRNAs to repress translation and initiate mRNA degradation [30]. MicroRNAs and lncRNAs have been implicated in AD pathogenesis, offering hope as potential biomarkers for disease diagnosis and the progression of epigenetic modifications. MiR-455-3p is an interesting example, as its overexpression results in a significant decrease in full-length Amyloid Precursor Protein (APP) and improved mitochondrial biogenesis, fusion, and synaptic genes, promoting cell survival and proliferation [31]. These findings emphasize the vital role of miR-455-3p in regulating abnormal APP processing, mitochondrial dynamics, and synaptic plasticity. Therefore, it is a promising molecule for further research in AD therapy.

Regarding lncRNAs and the involvement of NEAT1, an lncRNA, it has been reported that its downregulation increased p-tau generation, enhanced Aβ levels, and generated neuronal damage, accelerating the progression of AD [32]. On the other hand, individuals who have been diagnosed with AD have increased levels of Brain-Derived Neurotrophic Factor Antisense (BDNF-AS) in their bloodstream. These higher levels of BDNF-AS are strongly associated with cognitive abilities. In vitro assays have indicated that BDNF-AS can potentially cause neurotoxic effects. Additionally, the induction of BDNF-AS resulted in increased expression of BACE1 by inhibiting miR-9-5p. This, in turn, promoted the formation of amyloid plaques [33]. Among other significant elements in AD development are epigenetic modifications, as evidenced by numerous studies examining DNA methylation, histone modifications, and non-coding RNAs. For instance, cross-tissue analysis revealed hypermethylation of the *ANK1* gene in several brain regions of individuals with AD. In addition, differential methylation patterns were observed in genes associated with AD neuropathology, such as *ABCA7* and *BIN1*. Furthermore, histone tail modifications, including acetylation and methylation, were found to mediate chromatin condensation and regulate gene expression associated with synaptic function. These findings provide meaningful insights into the complex relationship between epigenetic mechanisms and AD pathology [34,35,36,37,38].

Mitochondrial trafficking to presynaptic terminals plays a crucial role in facilitating efficient synaptic vesicle release by regulating Ca^2+^ and ATP levels. However, pathological tau can disrupt this process by binding to kinesin and hindering the transport of mitochondria along microtubules. Consequently, this disruption may contribute to synaptic damage and eventual loss. Furthermore, pathological tau may also mislocalize to presynaptic terminals, where it can interfere with exocytosis by binding to synaptic vesicles [39,40] (risk factors are summarized in Figure 1).

Endolysosomal dysfunction in AD is a hallmark at early and preclinical stages; abnormalities in this system lead to impaired clearance of various proteins, including Aβ and tau. One of the primary consequences of endolysosomal dysfunction is the inefficient clearance of Aβ peptides. Disruptions in endolysosomal function can alter the processing of APP, leading to increased production or impaired clearance of Aβ, resulting in its accumulation and subsequent aggregation into amyloid plaques [41].

Moreover, endolysosomal dysfunction affects the turnover of other proteins involved in AD pathology, such as tau. The degradation of abnormal tau species is also regulated by the endolysosomal system, and dysfunction in this pathway can contribute to tau accumulation and pathology. Recent research has highlighted the role of genetic risk factors associated with endolysosomal function in AD, such as mutations in genes encoding proteins involved in endosomal trafficking and lysosomal function [42,43].

Additionally, environmental factors, including oxidative stress and inflammation, can disrupt endolysosomal function and exacerbate AD pathology. Furthermore, disruptions in endoplasmic reticulum (ER) homeostasis can precipitate ER stress, a condition implicated in the initiation and progression of AD pathology. ER stress ensues when the folding capacity of the ER is overwhelmed by the accumulation of misfolded or unfolded proteins [44]. Within the context of AD, diverse factors contribute to ER stress, notably the buildup of Aβ peptides, which directly provoke ER stress by perturbing calcium homeostasis and impeding protein folding. Cells mount a response to ER stress through the unfolded protein response (UPR), a signaling pathway aimed at restoring ER equilibrium by bolstering its folding capacity, degrading misfolded proteins, and curtailing protein synthesis. Nevertheless, persistent or severe ER stress can culminate in neuronal dysfunction and apoptosis, exacerbating neurodegeneration in AD [45].

Variants in the *PSEN1* gene also disrupt ER calcium homeostasis, further fueling ER stress. Dysregulated calcium signaling within the ER can precipitate mitochondrial dysfunction, oxidative stress, and ultimately neuronal apoptosis, thereby compounding the severity of AD pathology [46].

## 3. Pathological Mechanisms

Understanding the pathological mechanisms underlying AD is critical to the development of effective treatments and interventions. This section reviews the key pathological mechanisms, including the accumulation of Aβ plaques and neurofibrillary tangles, and examines how these mechanisms may differ between EOAD and LOAD [47].

The characteristic pathology of typical AD dementia includes the presence of extracellular amyloid plaques containing Aβ that are widely distributed throughout the cerebral cortex. Initially, Aβ accumulates in neocortical regions, followed by involvement of the limbic system, diencephalon, basal forebrain, and finally the cerebellum. In addition, inside neuron bodies, tau protein (which will be discussed in Section 3.2) undergoes abnormal chemical modifications (hyperphosphorylation), forming what are known as neurofibrillary tangles (NFTs). These tangles appear first in the medial temporal lobe and subsequently develop in isocortical regions of the temporal, parietal, and frontal lobes and interfere with normal cellular processes, disrupt neuronal function, and eventually, lead to cell death [48,49].

The receptors on the cell surface for ApoE are heparan sulfate proteoglycans (HSPGs), which serve as integral components in the orchestration of numerous intricate cellular mechanisms, including but not limited to cell adhesion, signaling cascades, and the effective clearance of Aβ peptides. Importantly, ApoE4 exhibits a binding affinity to HSPGs approximately three times higher than ApoE2 or ApoE3, indicating that the *APOEε4* allele, associated with AD risk, binds more strongly to HSPGs [50]. This interaction may affect the binding, uptake, clearance, and aggregation of Aβ peptides by neurons and microglia, leading to the accumulation of Aβ plaques in the brain, a hallmark of AD pathology [51].

Furthermore, the presence of ApoE4 may disrupt the normal function of HSPG in the brain, impairing synaptic plasticity and neuronal integrity. HSPG has been proposed to be critically involved in all phases of tau diffusion, hypothesized to mimic prion-like spread, including tau release from intracellular space to extracellular environment, cell surface binding, internalization, and aggregation. This disruption in the molecular interactions between HSPG and ApoE4 may contribute to the synaptic dysfunction, neuronal loss, and cognitive decline observed in AD (Figure 2 emphasize the central role of ApoE4 in AD development) [52].

### 3.1. Amyloid Plaques

The production of the Aβ peptide arises from the enzymatic processing of APP. APP is identified as a type I integral transmembrane glycoprotein characterized by a small intracellular C-terminal domain, an Aβ peptide region, and a large extracellular N-terminal domain [53]. APP undergoes proteolytic processing through two distinct pathways:Non-amyloidogenic pathway: In this pathway, APP is cleaved by α-secretases, such as ADAM10, resulting in the generation of a soluble fraction of alpha-APP (sAPPα) and a carboxy-terminal fragment (CTF-α). Subsequently, γ-secretase also acts on CTF-α, producing the P3 fragment. Importantly, P3 is a soluble peptide that lacks the propensity to aggregate, unlike Aβ [54].Amyloidogenic pathway: APP can also be cleaved at the β-site by a β-secretase (BACE1), resulting in the production of a soluble beta-amyloid precursor protein (sAPPβ) and a carboxy-terminal fragment composed of 99 amino acids (known as CTF-β or C99). Subsequently, γ-secretase acts on CTF-β, releasing the Aβ, which can vary in size (primarily Aβ40, Aβ42, and Aβ43), depending on enzymatic cleavage. The pathway described above leads to the accumulation of Aβ protein and, consequently, the formation of amyloid plaques. Although the 42-amino acid form (Aβ42) is suspected to be a causative agent in AD, the molecular basis of its neurotoxicity remains unknown. However, amyloid plaques can induce synaptic dysfunction and an inflammatory response that ultimately triggers neurodegeneration. In a healthy individual, the non-amyloidogenic pathway predominates, while in a patient with AD, Aβ clearance is lower than its production through the amyloidogenic pathway [55,56].

In EOAD, the presence of pathogenic variants in the *APP* gene can trigger an excessive production of Aβ protein, and variants in *PSEN1* and *PSEN2* may precipitate a reduction in γ-cleavage, fostering the production of longer and more toxic forms of Aβ, contributing to the onset of AD. Alternatively, in LOAD, the mechanisms underlying the accumulation of Aβ are more complex. While individuals carrying the *APOEε4* allele face an elevated risk of LOAD, it does not directly initiate the overproduction of Aβ. However, the presence of the *APOEε4* allele correlates with pronounced disruption in white matter integrity (Figure 3 summarize the pathological mechanism) [11,57].

### 3.2. Neurofibrillary Tangles

NFTs, consisting of aggregated tau protein, are another hallmark of AD pathology. Tau is a microtubule-associated protein that is typically found in the cytoplasm of axons, where it also plays a role in stabilizing microtubules, thereby supporting their structural integrity and essential cellular functions. However, it is also found in both presynaptic and postsynaptic areas and is associated with the nuclear envelope [58,59].

In AD and similar tau-related diseases, tau protein undergoes abnormal changes. It becomes excessively phosphorylated, leading to its detachment from microtubules and its assembly into paired helical filaments and straight filaments, which constitute the central component of NFTs. These accumulations of tau disrupt the structural stability of neurons and hinder intracellular transport processes [60,61].

Interestingly, recent investigations propose that the spread of tau pathology in AD may follow a mechanism akin to prions. In contrast to prion diseases, human tauopathies are not considered to be transmissible; however, the prion-like behavior of tau is still under study. The propagation process of tau protein likely involves multiple stages. Initially, tau aggregates, followed by its release from donor neurons. Subsequently, tau is taken up by specific recipient neurons, leading to the induction of tau aggregation within neurons, although the sequence of these events remains uncertain. Within this mechanism, the presence of abnormal tau aggregates in one neuron could trigger the alteration of normally folded tau proteins in nearby neurons, subsequently resulting in the dissemination of tau pathology across the entire brain. This spread phenomenon is thought to be a driving factor in the gradual progression of AD and the eventual involvement of different brain regions as the disease progresses [62,63,64,65,66].

The propagation of tau pathology frequently occurs via synapses, which are the junctions for communication between neurons. Anomalous tau aggregates have the ability to traverse from one neuron’s axon to the dendrites of a connected neuron, potentially triggering the formation of new tau aggregates within the receiving neuron. This mechanism could play a role in the dissemination of tau pathology within neural networks [67]. Recent research suggests that neurons have the ability to release tau aggregates into the extracellular environment. These tau aggregates in the extracellular space can subsequently be taken up by neighboring neurons or glial cells, either via endocytic pathways involving engulfment by the cell membrane, receptor-mediated or exosome-mediated mechanisms, or lipid raft-dependent mechanisms, among other proposed mechanisms. After internalization, tau undergoes intracellular trafficking where a portion is degraded through the endosome–lysosome route, but the remainder (non-degraded tau) can initialize pathological processes [68,69].

### 3.3. Differences of Pathological Mechanisms between EOAD and LOAD

The extent to which the neuropathologic features underlying EOAD differ from those of LOAD remains a subject of ongoing investigation, and certain differences have been observed. EOAD cases tend to have a higher burden of amyloid plaques and NFTs in the frontal cortex when compared to LOAD patients. In particular, the neocortex, specifically the parietal and occipital parietal regions, appears to bear a similar burden in both EOAD and LOAD cases [70].

In addition, individuals who develop AD at a younger age often have a neuropathological profile characterized by a “purer” AD pathology (plaques and tangles), with fewer concurrent neuropathological changes compared to older AD patients. This can be explained in great proportion by the intervention of genetic factors, which have a stronger effect or as the primary cause of the disease in younger cases, and the young age, at which comorbidities are less frequent. Older AD patients often have a range of coexisting pathologies alongside the typical AD inclusions, including Lewy bodies and vascular pathology. Tau pathology primarily initiates within the entorhinal cortex and hippocampus, leading to early-stage neurofibrillary degeneration, synaptic and neuronal loss, as well as regional atrophy. Subsequently, in the early to intermediate stages of the disease, tau pathology extends to affect the locus coeruleus, basal forebrain, and associated neocortical regions, culminating with the involvement of the primary sensory cortex [9,13,71,72].

## 4. Diagnosis and Assessment

Accurate and timely diagnosis of AD is critical for effective management and intervention. This section provides a comprehensive comparison of the diagnostic and assessment strategies used in both EOAD and LOAD, taking into account various factors, including neuroimaging techniques, biomarkers, and cognitive assessments [73,74].

### 4.1. Neuroimaging

Neuroimaging plays a central role in the diagnosis and understanding of AD, providing valuable insight into the structural and functional changes that occur in the brain. While there is considerable overlap in the use of neuroimaging techniques for EOAD and LOAD, there are subtle differences that merit exploration [75].

#### 4.1.1. Positron Emission Tomography (PET)

PET imaging has revolutionized our ability to visualize molecular and metabolic processes in the brain, making it an invaluable tool in the diagnosis and characterization of AD. Here, we take a closer look at how PET imaging is used in EOAD and LOAD [76].

Amyloid PET imaging uses radiotracers such as Pittsburgh compound B (PiB) and florbetapit (18F-AV-45) to detect the presence and distribution of amyloid plaques in the brain. This technique is particularly relevant in EOAD and LOAD, as the accumulation of AB is a hallmark. However, the extent and pattern of amyloid deposition can vary. As NFTs are also a hallmark of AD, this technique can provide valuable insight into the distribution and severity of tau pathology [77,78].

EOAD: These patients often have more aggressive and widespread deposition of amyloid plaques, which are easier to detect with PiB PET scans. The distinctive amyloid burden in EOAD cases helps distinguish it from other neurodegenerative diseases. They also have more extensive and severe tau pathology, and tau PET scans can show the widespread distribution of tau aggregates in the brain. This information helps to confirm the diagnosis and assess disease progression [79].

LOAD: These patients also benefit from amyloid PET scans, especially when the diagnosis is difficult due to comorbidities or overlapping symptoms with other dementias. While LOAD generally presents with amyloid plaques, the pattern may be less aggressive than in EOAD cases. Tau PET imaging can also be useful in LOAD, particularly when there is a need to differentiate AD from other neurodegenerative conditions that may exhibit different patterns of tau pathology [80,81,82].

#### 4.1.2. Magnetic Resonance Imaging (MRI)

MRI is a versatile imaging modality that provides detailed structural information about the brain. MRI is instrumental in the diagnosis and characterization of EOAD and LOAD, providing valuable insight into the extent and pattern of structural abnormalities [83].

In EOAD, MRI often shows prominent structural changes. These changes may include significant cortical atrophy, particularly in the frontal and temporal lobes, which are commonly affected in this subtype. The degree of atrophy observed in EOAD may be more severe and widespread than in LOAD. MRI can precisely delineate these structural changes, helping to differentiate AD from other neurodegenerative diseases [84,85].

LOAD also has structural abnormalities that can be detected by MRI. Although the pattern of atrophy may be different from EOAD, MRI remains a valuable tool for assessing regional brain atrophy. The medial temporal lobe, including the hippocampus, is often one of the first areas affected in LOAD, and MRI can provide detailed images of these changes. In addition, LOAD may have more widespread cortical thinning, which can be detected by MRI [85].

### 4.2. Cognitive Assessment

Cognitive or neuropsychological assessment for AD is essentially the same as those used for dementia in general. It involves tests and evaluations designed to assess various aspects of cognitive function (attention, memory, language, visuospatial skills, executive function) to determine the patient’s capacity and monitor their progression over time [86].

Among the most commonly used is the Mini-Mental State Examination (MMSE), which is the most widely used and validated neuropsychological tool for assessing an individual’s cognitive functional status [87,88]. It consists of a 30-point questionnaire, in which scores of 24 or below indicate cognitive impairment, used to assess spatial–temporal orientation, short-term verbal memory, delayed recall, linguistic ability, numerical calculation, and construction of simple figures in individuals suspected of having a neuropsychological problem [86]. Another neuropsychological tool is the Clinical Dementia Rating (CDR) [89]. It is a semi-structured interview conducted with the patient (affected or suspected to be) and their caregiver (informant). The questions are directed to determine whether the patient exhibits memory problems, judgment issues, difficulty with problem solving, daily and social activities, etc., from the informant and patient point of view. The results can range from 0 to 3: no dementia (CDR = 0) to severe cognitive impairment (CDR = 3). Although this test can discriminate between mild cognitive impairment and a patient without impairment [90], the CDR is biased in identifying early-stage dementia [91]. A final example would be the Clock Drawing Test (CDT). This test includes activities aimed to assess attention, mathematical ability, comprehension, construction, naming, orientation, recall, repetition, spelling, and writing, all necessary skills for drawing a clock, including the figure outline, clock hands, and determining the hours [86].

All cognitive assessment must be conducted by trained healthcare professionals like neurologists, neuropsychologists, or geriatricians, and interpreted in the context of the individual’s medical history, clinical symptoms, other diagnostic tests (e.g., neuroimaging), educational achievement, and standardized scores for specific populations [86].

### 4.3. Biomarkers

Biomarkers in AD have significant diagnostic and prognostic utility, providing valuable information for early diagnosis and prediction of disease progression. Understanding their utility is essential for improved patient care and the development of targeted therapies. Current AD biomarkers include identification, in cerebrospinal fluid (CSF), of pathogenic hallmarks Aβ40, Aβ42, total-tau and p-tau, neurodegeneration markers (neurofilament light (NFL), and inflammation markers like interleukin 1α (IL-1α), IL-1β, IL-6, IL-8, IL-33, intercellular adhesion molecule 1 (ICAM-1), progranulin, SDF-1, soluble interleukin 1 receptor-like (sST2), and vascular cell adhesion protein 1 (VCAM-1). For EOAD cases, genetic biomarkers such as *APP*, *PSEN1*, and *PSEN2* mutations are added to the list. The existence of mutations on one of these genes changes the expected values of certain biomarkers. For example, it has been described that *PSEN1* mutation carriers show decreased Aβ40 levels and increased Aβ42/Aβ40 ratio levels, while *PSEN2* mutation carriers show increased Aβ42 levels and increased Aβ42/Aβ40 ratio levels. Nevertheless, biomarker profiles between EOAD and LOAD, through biomarker analyses only, seem far from being reached [92]. Emerging biomarkers in the field of AD research offer promising avenues for early diagnosis, disease monitoring, and potential therapeutic interventions [93].

The need to expand the range of cerebrospinal fluid (CSF) biomarkers used to characterize different aspects of AD pathophysiology is underscored by the diversity of pathological presentations observed in LOAD [94]. The inclusion of neurogranin, a synaptic protein with AD specificity and predictive value for the future rate of cognitive decline, is a potential and promising addition to the AD CSF biomarker toolkit [95,96].

While CSF- and PET-based assessments are available, their use remains limited due to limitations such as high cost and perceived invasiveness [97]. However, rapid advances in the development of highly sensitive assays have opened the possibility of measuring levels of pathological brain-related and AD-associated proteins in the blood, and recent research has shown encouraging results in this regard. Among these, plasma p-tau appears to be emerging as a prime candidate marker, particularly in cases of symptomatic AD, including prodromal AD and AD dementia, when combined with the Aβ42/40 ratio. Although not an exclusive indicator of AD, blood neurofilament light chain (NfL) is a promising marker of neurodegeneration and could serve as a tool for monitoring the effect of disease-modifying therapies [93,98,99].

## 5. Treatments and Therapeutic Approaches

Several treatment options have been developed and new therapeutic approaches are being explored as research into the disease continues. The choice of appropriate therapy may vary depending on the onset of the disease, whether EOAD or LOAD.

### 5.1. Conventional Treatments

#### 5.1.1. Cholinesterase Inhibitors

Cholinesterase inhibitors (CIs) are a class of drugs commonly used in the treatment of AD. These drugs play a critical role in managing the cognitive and behavioral symptoms of AD, especially in its early and moderate stages. CI works by increasing acetylcholine levels in the brain [100]. Acetylcholine is a neurotransmitter that plays a key role in cognitive functions such as memory, learning, and decision making. In AD, acetylcholine is reduced due to excessive breakdown by the enzyme cholinesterase. Cholinergic neurons are a type of nerve cell that utilizes acetylcholine as its primary neurotransmitter. These neurons are found in several regions of the brain and the peripheral nervous system. In AD, there is a selective degeneration or death of these cholinergic neurons [100,101].

CI may help improve memory, attention, and other cognitive functions in patients with AD. While these effects do not reverse the disease, they can lead to improved quality of life and greater functional independence. In addition to their cognitive effects, cholinesterase inhibitors may also help reduce behavioral and psychological symptoms associated with AD, such as agitation, depression, and aggression [102].

The most commonly used CIs in the treatment of AD include donepezil, rivastigmine, and galantamine. These drugs have slightly different efficacy and side effect profiles, allowing physicians to choose the most appropriate drug for each patient [103,104].

CIs are typically used in the early and moderate stages of AD, when cognitive symptoms are most pronounced. These drugs have been shown to be most effective in these stages, as they work to preserve the remaining cognitive capacity and slow its decline. In the advanced stages of AD, cholinesterase inhibitors may become less effective as the disease progresses and behavioral and physical symptoms become more pronounced. Donepezil has been approved to treat all AD stages, while galantamine and rivastigmine have been approved for mild and moderate stages. However, in some cases, these drugs may still provide benefit in symptom management [105,106,107].

Although CIs are generally safe, they can cause side effects such as nausea, vomiting, and diarrhea. Doctors closely monitor patients taking these drugs to minimize these effects [108].

#### 5.1.2. NMDA Receptor Inhibitor

Memantine, a pharmacological agent utilized in the management of AD, falls under the category of NMDA receptor antagonists. Its mechanism of action revolves around modulating the activity of glutamate, a neurotransmitter crucial in memory and learning processes; normally, NMDA receptors are activated by glutamate, leading to the influx of calcium ions into the postsynaptic neuron. This calcium influx is crucial for synaptic plasticity and memory formation. In the context of AD, glutamate activity becomes dysregulated, and Aβ oligomers can bind and activate NMDA receptors. However, excessive or dysregulated activation of NMDA receptors, such as that caused by Aβ oligomers, can lead to excitotoxicity, neuronal damage, and ultimately cell death [109,110].

The chronic activation of NMDA receptors by Aβ oligomers disrupts the delicate balance of synaptic transmission and plasticity, contributing to the cognitive impairments observed in AD. Understanding this mechanism is important for developing potential therapies aimed at mitigating the toxic effects of Aβ oligomers on synaptic function and preserving cognitive function. By interfering with NMDA receptors, memantine helps restore glutamate balance, mitigating the excitotoxic repercussions associated with excessive stimulation. This action serves to safeguard nerve cells from damage and decelerate the progression of AD symptoms. It is worth highlighting that memantine is typically administered for moderate to severe AD cases, often complemented with other drugs like CI. While memantine exhibits potential in enhancing cognitive function and delaying symptom exacerbation in some patients, its efficacy varies among individuals, necessitating vigilant monitoring by healthcare providers (the mechanisms of action of both CI and memantine are summarized in Figure 4) [109,110].

### 5.2. Emerging Treatments

#### 5.2.1. Monoclonal Antibodies

The development of safe and effective drugs for treating AD has always been the goal. Currently, anti-amyloid therapies are showing promising outcomes. Monoclonal antibodies (mAbs) against tau proteins are designed to bind to extracellular tau proteins. This action slows down or prevents their distribution between cells, inhibiting tau protein aggregation and the formation of neurofibrillary tangles. Additionally, these antibodies help regulate the clearance and production of Aβ. Several anti-Aβ monoclonal antibodies such as Solanezumab, Bapineuzumab, Gantenerumab, Aducanumab, Lecanemab, and Donanemab are currently undergoing randomized clinical trials (RCTs) with varying results. Solanezumab was among the first anti-Aβ monoclonal antibodies to be tested in preclinical and clinical AD studies. It was found that solanezumab reduced brain Aβ levels by targeting soluble monomeric forms of Aβ peptide but had little effect on deposits. However, this compound, unfortunately, accelerated cognitive decline in both asymptomatic and symptomatic trial participants [111,112].

In contrast, Aducanumab captures aggregated Aβ and indirectly supports clearance of Aβ, most likely by presenting it to microglia for phagocytosis, thereby increasing the clearance of Aβ [113]. Results from the EMERGE Phase III clinical trial in EOAD patients treated with high-dose Aducanumab showed a significant reduction in clinical decline of CDR-SB scores at 78 weeks compared to baseline [114]. Individuals who received Aducanumab showed reduced clinical decline and had significant cognitive and functional benefits, such as improved orientation, language, and memory [115]. Lecanemab, a mAB against Aβ, is in phase III trials. It has been shown to reduce markers of amyloid in EOAD and resulted in a moderate decline in measures of cognition and function [116].

Studies have shown that Donanemab has the most significant impact on slowing down the progression of tau in individuals who have completely cleared amyloid, specifically in the brain regions that are in the advanced stages of the disease. It is important to further study these findings to understand how individuals with APOEε4 status respond to this treatment [117].

Emerging therapeutic approaches for AD have focused on leveraging the immune system to target Aβ (Figure 5). Among these strategies, activating antibodies targeting TREM2 and the Syncytin-1 protein (SYNK) has shown promise. TREM2 is expressed on microglia and is involved in the regulation of the immune response in the brain. Activation of TREM2 can enhance microglial phagocytosis and clearance of Aβ plaques. Similarly, SYNC, a protein involved in syncytin-mediated cell–cell fusion, has been identified as a potential target for antibody therapy due to its role in Aβ degradation. Conversely, inhibitory antibodies targeting Cluster of Differentiation 33 (CD33) and Src homology region 2 domain-containing phosphatase-1 (SHP1) aim to dampen the inflammatory response and promote Aβ clearance. CD33 is expressed on microglia and acts as an inhibitory receptor, while SHP1 negatively regulates immune signaling pathways. By blocking these inhibitory pathways, the goal is to enhance microglial activity and facilitate the removal of Aβ deposits. On one hand, microglial activation can be beneficial, as it contributes to the clearance of Aβ plaques, as activated microglia are capable of phagocytosing and clearing Aβ from the brain. This process is crucial for reducing the burden of Aβ oligomers, and microglia can also recognize and engulf Aβ through various receptors. On the other hand, prolonged or dysregulated microglial activation can lead to a state of chronic neuroinflammation. In this state, microglia release pro-inflammatory cytokines, chemokines, and reactive oxygen species, which can damage neurons and exacerbate neurodegeneration. Therefore, novel immunotherapeutic approaches offer promising avenues for the treatment of AD by harnessing the body’s immune system to target and degrade pathological protein aggregates implicated in disease progression [118,119].

#### 5.2.2. Hybrid Molecules with Dual Affinity

Hybrid molecules with dual affinity represent a novel approach in drug design aimed at addressing complex neurological and psychiatric disorders. By targeting two receptors simultaneously, these hybrid molecules aim to modulate two distinct but interconnected pathways implicated in conditions such as AD. The rationale behind this approach lies in the potential synergistic effects of targeting multiple molecular targets involved in the pathophysiology of these complex disorders. The reliance on highly selective drugs targeting specific aspects of AD pathology has proven not so effective, necessitating the use of combination therapies to address the array of symptoms associated with disease progression [120].

An effective therapeutic strategy for AD may hinge on the ability of newly developed agents to interact with and regulate multiple molecular targets, thereby concurrently modulating numerous interconnected pathogenic pathways.

Within this realm, numerous compounds with multitarget profiles have emerged, demonstrating intriguing and promising pharmacological characteristics that position them as prospective therapeutic candidates. Hybrid therapeutic complexes, developed through specific combinations of two bioactive pharmacophores, yield homo- and heterodimers with enhanced affinity, therapeutic efficacy, biological profile, and complementary effects. Consequently, many disease-modifying hybrids hold promise for advancement into next-generation AD medicines (examples are shown in Figure 6).

Designing hybrid multitargeted therapeutic compounds represents a promising approach in the pursuit of effective AD treatments. However, several challenges must be addressed, and further research is warranted to develop hybrid therapeutic compounds suitable for clinical use while mitigating off-target adverse effects [121,122,123].

#### 5.2.3. Gamma-Secretase Modulators

Gamma-secretase modulators (GSMs) regulate the action of gamma-secretase enzymes, thereby reducing the production of Aβ. It has been observed that the ratio of Aβ37 or Aβ38 to Aβ42 or Aβ43 levels in CSF has a correlation with AoO as well as scores on the MMSE in individuals with fAD mutations [124]. There is also a link between the proportion of shorter peptides (Aβ37 + 38 + 40) and AoO in familial fAD caused by different *PSEN1* mutations [125].

Pfizer conducted a promising phase 1 trial with a drug referred to as PF-06648671, which showed reductions in Aβ42 and Aβ40 levels, along with increases in Aβ37 levels, in healthy volunteers [126]. Therefore, Gamma-secretase modulators do not affect the overall production of Aβ peptides but instead modify the balance between longer and shorter Aβ forms. Thus, the potential signaling role of Aβ in the presence of GSMs may remain intact. As discussed above, the initial increase in Aβ pathology, detected through PET scans, marks the first pathological alteration in AD, which subsequently triggers tau pathology. Aβ peptides have significant physiological functions in normal cellular processes. Thus, using a GSM before the rapid escalation of tau pathology driven by Aβ could present an alternative strategy for treatment for the early stages of AD [127,128].

## 6. The Effect of ApoE

The gene for *APOE*, located at 19q13.32, encoding for ApoE in humans, is a 299-amino acid glycoprotein with a molecular weight of 34 kDa, primarily produced by astrocytes and activated microglia in the brain and hepatocytes in the liver. ApoE is composed of two primary structural segments linked by a flexible region. The first segment, spanning residues 1 to 167, encompasses the area responsible for receptor binding. Conversely, the second segment, spanning residues 206 to 299, houses the region responsible for lipid binding. ApoE plays a crucial role in lipid metabolism and transport, particularly in the central nervous system (CNS). In the brain, ApoE facilitates the transport of cholesterol and other lipids to neurons through interactions with cell surface receptors and also plays a key role in several brain functions, including lipid metabolism, processing of Aβ peptides, neural growth, and immunoregulation. ApoE exists in three main isoforms: ApoE2, ApoE3, and ApoE4, which are encoded by different alleles of the *APOE* gene. The structural differences between these isoforms arise from amino acid substitutions at positions 112 and 158. ApoE2 has cysteine residues at both positions, ApoE3 has a cysteine at position 112 and an arginine at position 158, while ApoE4 has arginine residues at both positions [129,130]. In the context of AD, the *APOEε4* allele is a well-studied risk factor for developing the disease and other dementia [131]. On the other hand, *APOEε2* is considered a protective allele [132], while *APOEε3* is considered neutral in the risk of developing AD, not considering *APOEε3* rare variants like Christchurch (*APOE*-*R136S*), which exerts a protective effect [133]. Nonetheless, this risk is relative, changing according to population characteristics. It has been described that the risk conferred by the *APOEε4* allele is lower on Latin American, African, and other non-white populations due to ethnic origin in comparison to Caucasians [134].

Besides the well-studied effect of ApoE on physiological matters, it is debated whether it may also modulate the cognitive and structural phenotype of the disease and if its effect changes relative to LOAD or EOAD.

### 6.1. Effects of ApoE on LOAD

#### 6.1.1. Effect of ApoE on Cognition

In previous research, it has been described that the effect of *APOEε4* on the LOAD population reduces the age of symptom onset [132], increases disease progression, impairs cognitive function, especially episodic memory [135], is related to an increase in the regional density of Aβ and tau aggregates [136], decreases hippocampal volume [135], increases levels of biomarkers in cerebrospinal fluid, and decreases glucose metabolism [137], among other effects.

Morgen et al. 2013 [138], through neuroimaging techniques, showed that the presence of *APOEε4* was related to a decrease in temporal medial lobe volume, a crucial structure for memory processing, while *APOEε4* non-carriers showed more cortically generalized damage. This structural damage of the temporal medial lobe, for *APOEε4* carriers, is reflected in results of neuropsychological research showing the appearance of amnestic presentations, meaning that memory is the main cognitive domain affected, that would account for typical AD presentations. On the other hand, non-carriers showed lower performances in executive functioning evaluation (trail making test, part B) and cognitive function related to cortical and prefrontal areas. On the contrary, the absence of the *APOEε4* allele in LOAD patients was related to slow progression of cognitive decline, decreasing 1.3 points per year on MMSE in comparison with EOAD patients (discussed below) [139,140,141].

#### 6.1.2. Effect of ApoE on Aβ

The differential impact of ApoE isoforms on AD risk involves ApoE4’s detrimental effect and ApoE2’s protective role regarding Aβ severity, plaque burden, and cerebral amyloid angiopathy (CAA) [142]. The tertiary conformation differences across ApoE isoforms affect their interaction affinity with Aβ and propensity for enzymatic cleavage, potentially generating toxic fragments. The interaction between ApoE and pathological Aβ deposition is central to AD risk, influencing plaque morphology and fibril formation [143,144]. ApoE also plays a role in Aβ clearance via receptor-mediated mechanisms, with LRP1 receptors facilitating Aβ/ApoE complex uptake. However, the less-effective lipid transport of ApoE4 compromises Aβ clearance, contributing to reduced Aβ elimination [145]. These findings suggest that targeting the interaction between ApoE and Aβ may offer therapeutic intervention opportunities at early disease stages, emphasizing the importance of understanding the differential effects of ApoE isoforms on AD pathogenesis (Figure 6) [146].

#### 6.1.3. Effect of ApoE on Neuroinflammation

Microglial function appears to be influenced by isoform-specific effects, particularly regarding the response to Aβ pathology. Recent findings indicate that ApoE3 is more effective than ApoE4 in stimulating microglial responses to Aβ injection, potentially mediated by the triggering TREM2 [147]. TREM2, expressed by microglia in the brain, exhibits a high affinity for ApoE and modulates microglial responses, suggesting their involvement in chemotaxis towards plaques [148]. These insights underscore the intricate interplay between ApoE, TREM2, and microglial responses in the context of AD pathology, highlighting potential avenues for therapeutic targeting to enhance neuroprotection. sTREM2 is a soluble form of the TREM2 receptor that is released into the extracellular space. Although the exact function of sTREM2 is not completely clear, it is known to be involved in several cellular processes in astrocytes, neurons, and microglia [149].

ApoE4 leads to reduced levels of ApoE in both microglia and astrocytes. This reduction in ApoE levels makes the brain more susceptible to inflammation, which is indicated by higher levels of TNFα even under resting conditions. ApoE plays a crucial role in modulating microglial response to inflammation. Microglia can release TNFα, with increased ApoE secretion observed in ApoE2 and ApoE3 but not in ApoE4 microglia. In astrocytes, inflammatory stimuli leave ApoE2 and ApoE3 levels unaffected, while ApoE4 is diminished [150,151,152].

#### 6.1.4. Effect of APOE on Blood–Brain Barrier

APOE has been linked to a more severe CAA, resulting in a heightened risk of lobar intracerebral hemorrhage, as well as other cerebrovascular complications. ApoE4 has been found to independently increase BBB permeability compared to ApoE3, as demonstrated in knock-in mouse models and confirmed through dynamic contrast-enhanced MRI in individuals with different cognitive statuses [153,154]. Proposed mechanisms underlying this effect include the activation of cyclophilin A, leading to elevated levels of MMP9 and pericyte injury, as well as disruption of the capillary basement membrane [155]. Mounting evidence suggests that BBB breakdown could be one of the earliest events in the pathogenesis of cognitive decline and AD pathology [156]. ApoE has been implicated in modulating cerebrovascular tight junction integrity, independent of CAA presence in AD brains. Additionally, ApoE4 disrupts endothelial BBB integrity by influencing various cellular processes, including extracellular matrix regulation, cell adhesion, cytoskeleton stability, and translation in brain endothelium. Moreover, studies have indicated that BBB breakdown contributes to cognitive decline in *APOEε4* carriers, irrespective of Aβ or tau pathology [157,158,159]. Vascular endothelial cells, forming tight junctions to create a functional barrier, are a crucial component of the BBB. The progressive BBB breakdown associated with ApoE4 presence leads to synaptic dysfunction and behavioral deficits. Dysfunction in pericytes, which regulate the BBB by influencing gene expression in endothelial cells and inducing polarization of astrocytic end-feet, further contributes to BBB impairment [160]. Pericytes expressing ApoE4 exhibit reduced capacity to support endothelial cell function, thereby compromising BBB integrity and heightening susceptibility to cognitive decline [153].

#### 6.1.5. Effect of ApoE on Tau

ApoE’s role in influencing tau neuropathological changes in AD brains is multifaceted. While ApoE3 may inhibit abnormal hyperphosphorylation and destabilization of the neuronal cytoskeleton in AD, the C-terminal-truncated form of ApoE4 is neurotoxic and stimulates tau phosphorylation, contributing to pre-NFTs [161,162]. Although there is little direct interaction between ApoE and tau in vivo, studies in transgenic mice suggest that ApoE isoforms indirectly affect tau pathology through their influence on microglia phenotype, with ApoE4 promoting tau-induced neurodegeneration compared to ApoE3 [163,164]. LRP1 has been identified as a receptor for tau uptake by neurons, and ApoE affects tau’s ability to bind LRP1, potentially influencing tau intracellular trafficking in an isoform-dependent manner [165]. The involvement of different receptors in tau uptake across various cell types remains unclear, particularly regarding how tau escapes to the cytoplasm for interaction with endogenous tau or accumulation as glial fibrillary tangles. Nonetheless, ApoE likely influences tau intracellular trafficking in an isoform-specific manner, suggesting a complex interplay between ApoE and tau pathology (the effects of ApoE are summarized in Figure 7) [166].

### 6.2. Effect of ApoE on EOAD

As much as it is clear that *APOEε4* is a risk factor, it is not fully understood whether it represents a risk factor for EOAD cases, mainly because the literature on its effect over AoO, clinical features, and cognitive phenotype is not as extensive as for LOAD cases. In general, a lower age at onset (EOAD cases) seems to be associated with faster progression of the disease [140]. However, articles show great heterogeneity in their results. Regarding AoO, De Luca et al., 2016 and Valdez-Gaxiola et al., 2023 [167,168] showed that the presence of *APOEε4* retards the appearance of EOAD symptoms, while studies like Langella et al., 2023 [169] suggest that it decreases AoO and accelerates cognitive decline in *PSEN1* E280A carriers (related to fAD). Nonetheless, Ryan et al., 2016 [170], studying a fAD population, found no effect of *APOEε4* over AoO nor cognitive symptoms.

About ApoE genotype over cognition, the study by Van der vlies et al., 2007 [171] showed more rapid global cognitive decline in early-onset patients, especially prominent in *APOEε4*-negative patients. Smits et al., 2015 [172] went further, evaluating all cognitive functions, to discover the effect of the allele on each. They observed that younger AD patients, especially when they are *APOEε4*-negative, show faster decline in language, attention, executive and visuo-spatial functioning, all non-memory domains. The majority of *APOEε4*-positive EOAD patients had worse results for memory tasks than non-carriers of the *APOEε4* allele, while *APOEε4*-negative AD patients declined fastest on non-amnestic domains, which is congruent with amnestic and non-amnestic phenotypes seen in LOAD [173,174]. Recent results from a Swedish population carrying *PSEN1* and *APP* variants related to fAD showed that *APOEε4* did not have an effect as a main predictor over cognition, but it did when interacting with other factors. Episodic memory was significantly affected by the interaction between *APOEε4* and *APP/PSEN1* mutations, showing favorable performance in the absence of *APOEε4* in *PSEN1* compared to *APP* mutation carriers [175,176]. Also, interactions between *APOEε4* and AoO showed differential effects depending on whether it was *APP* or *PSEN1* mutation carriers; in the first group, verbal abilities and attention were maintained in comparison with those who lacked the *APOEε4* allele. In the latter, a detrimental effect of the *APOEε4*–AoO interaction was observed in visuospatial and verbal abilities [175].

Regarding global cognitive decline, Van der vlies et al. 2007 [141] showed that EOAD patients without the *APOEε4* allele had rapid disease progression, losing approximately 2.4 MMSE points per year (in comparison with the 1.3 points lost in LOAD cases). This demonstrates the existence of an opposite effect of *APOEε4* over cognition between EOAD and LOAD patients. Authors have addressed this heterogeneity in disease progression, rate of cognitive decline, and the cognitive domains that are affected in patients with early-onset versus late-onset AD, as well as epistasis or other complex genetic interactions when EOAD is caused by pathogenic mutations [175,176], antagonist pleiotropic effect of *APOEε4* heterozygosity over the adult life course [177], the modulation of educational attainment [169], or the interaction of environmental factors yet to be discovered. Meanwhile, this heterogeneity in results means that the effect of *APOEε4* cannot be or must not be generalized when talking about EOAD. Focusing on ApoE and its differential impact on EOAD and LOAD provides an opportunity to unravel the complex genetic and molecular mechanisms underlying AD. This knowledge is essential for developing effective therapeutic interventions that address the specific pathological processes associated with each disease subtype (All the differences between LOAD and EOAD are summarized in Table 1).

## 7. Future Perspectives

By 2025, Alzheimer disease will affect more than 150 million people, placing increasing strain on healthcare systems and caregivers. That is why it is crucial to understand risk factors, disease progression, and potential interventions. In the above, we provide a comprehensive review of the majority of points that encompass the AD umbrella, offering insights into the unique challenges and considerations of its traditional classification according to AoO. Regarding the information available on the disease, we highlight the need for increased awareness, not only in research, but also in social contexts, and resources to support affected individuals and their families, especially for EOAD, which tends to receive less attention in terms of public discourse and awareness campaigns. AD is often associated with older age, and the focus may disproportionately center on the more common late-onset form. However, in the research field, most of the experimental models are based on EOAD. This can result in less visibility and advocacy for individuals and families affected by EOAD compared to LOAD, yet its impact can be equally devastating, considering that it affects adults in their midlife, a period of time in which they can become parents, achieve professional goals, and find an economic balance, among other things.

Regarding diagnosis, proper patient classification will provide appropriate and on-time diagnosis, treatment, interventions, and support. Nevertheless, the latter tends to be more difficult for LOAD cases due to the multifactorial origin, and a not-so-clear preclinical stage. Emerging biomarkers in the field of AD research offer promising avenues for early diagnosis, disease monitoring, and potential therapeutic interventions. The need to expand the range of CSF biomarkers used to characterize different aspects of AD pathophysiology is underscored by the diversity of pathological presentations observed in LOAD. The inclusion of neurogranin is a potential and promising addition to the AD CSF biomarker toolkit. Advocacy efforts may focus on a more multifaceted approach. This includes genetic and medical analysis, genetic counseling, psychological accompaniment, and the caregiver perspective. This integrative approach encompasses not only the medical part of the disease, but also its personal and social implications. Emerging therapeutic approaches focus on leveraging the immune system to target Aβ. Among these strategies, activating antibodies targeting TREM2 and SYNK has shown promising results. Furthermore, given the broad impact of ApoE on various systems and parts of the body, it may exert its influence or have effects in a variety of ways, manifesting itself in multiple physiological functions and biological processes. While the *APOEε4* allele is the primary risk factor for AD, finding a way to alter the mechanism of action of ApoE or its receptors could be an effective treatment for the prevention and/or treatment of AD. However, considering the inverse effects observed in EOAD and LOAD, it may be necessary to first identify the cause.

Additionally, another promising avenue for therapy involves altering the structure of ApoE through genetic engineering or small molecule intervention. The CRISPR/Cas9 system holds promise for directly converting ApoE4 to ApoE3 or ApoE2. Similarly, an AAV system could be employed to stimulate the expression of ApoE2. Alternatively, small molecule inhibitors could be utilized to disrupt interdomain interactions, leading to structural modifications of ApoE and alterations in its functionality, leading to a better clearance of Aβ [179]. Pharmaceutical therapies can be complemented by non-pharmacological interventions. For example, new research is focusing on the potential benefits of nutraceuticals, fortified foods with health-promoting ingredients like curcumin, omega-3 fatty acids, flavonoids, and certain vitamins, microbiota interactions, and fasting, which have shown promise in preclinical studies and early clinical trials for their potential to mitigate AD pathology or symptoms. Also, other lifestyle modifications, including exercise, cognitive stimulation, and social engagement, are gaining popularity and are being consolidated as important components of the management of AD [181].

As science continues to extend the average human lifespan, the prevalence of AD is expected to increase, and so typical classification parameters such as AoO will definitely change. Improving access to specialized care worldwide, promoting and funding research, from basic to specialized, making efforts to destigmatize the condition through scientific diffusion, proper recruiting of patients for clinical trials, creation of public policies, and social initiatives, on the whole, will help ensure earlier diagnosis and interventions to enhance the quality of life for both patients and their families.

## 8. Conclusions

AD is a major global health challenge due to its complex etiology. The conventional understanding of AD may oversimplify its nature, especially concerning late-onset cases. LOAD is influenced by a myriad of factors, including genetics, environment, medical conditions, and lifestyle choices, rendering it difficult to classify as a singular pathological entity. Recent studies focusing on Aβ have highlighted the limitations of a uni-dimensional therapeutic approach, highlighting the necessity for a holistic strategy to effectively prevent and manage AD. Central to this approach is a thorough understanding of the heterogeneous nature of AD and the intricate interplay of different underlying pathological mechanisms. In addition, it is imperative to recognize the potential impact of protective factors on the clinical manifestation of the disease. While significant progress has been made in unraveling the molecular and genetic basis of AD, numerous questions, particularly regarding EOAD and its distinct genetic and clinical features, remain unanswered. Filling these knowledge gaps will require collaborative endeavors and the inclusion of diverse populations in research endeavors. Looking ahead, the prospect of personalized medicine, guided by comprehensive genetic and clinical profiling, holds immense potential in devising more efficacious strategies for the prevention and treatment of AD.

## Figures and Tables

**Figure 1 diseases-12-00110-f001:**
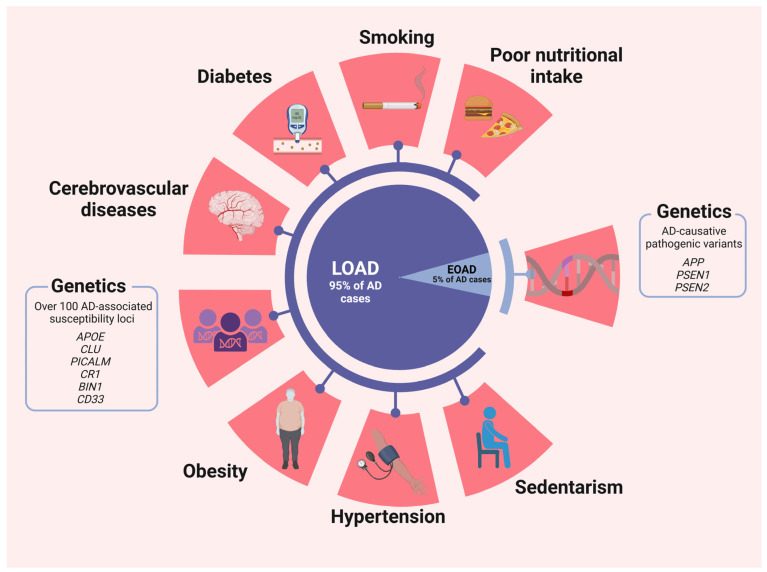
Risk factors for LOAD and EOAD. Although some risk factors have a more prominent effect in one subtype than the other, it is important to clarify that they are not associated uniquely to one or another. Created with BioRender.com.

**Figure 2 diseases-12-00110-f002:**
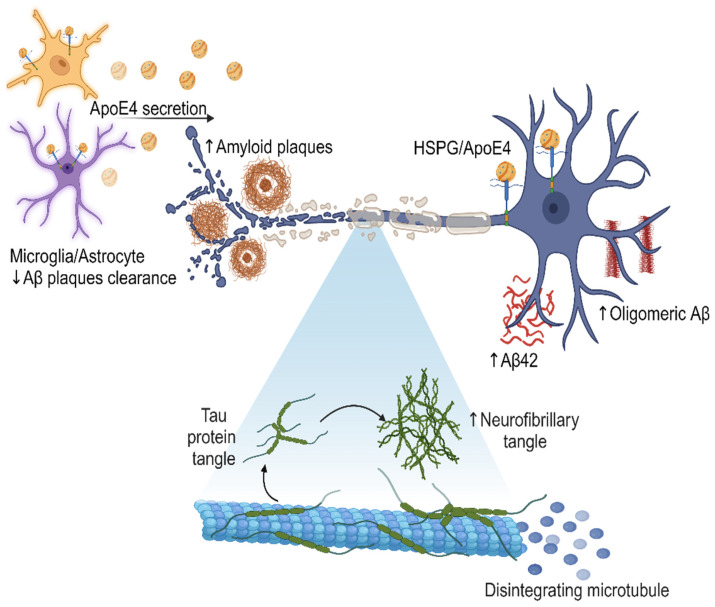
Secretion of ApoE4 by microglia and astrocytes in the brain, leading to an increase in the amount of Aβ, oligomeric Aβ, and amyloid plaques. The presence of ApoE4 also correlates with the formation of neurofibrillary tangles, which disintegrate microtubules and contribute to neurodegeneration. The effect of the interaction between ApoE4 and HSPG may further modulate amyloid pathology and disease progression. This visual representation illustrates the molecular mechanisms and cellular interactions involved in the AD pathological cascade, highlighting the central role of ApoE4 in this process. Created with BioRender.com.

**Figure 3 diseases-12-00110-f003:**
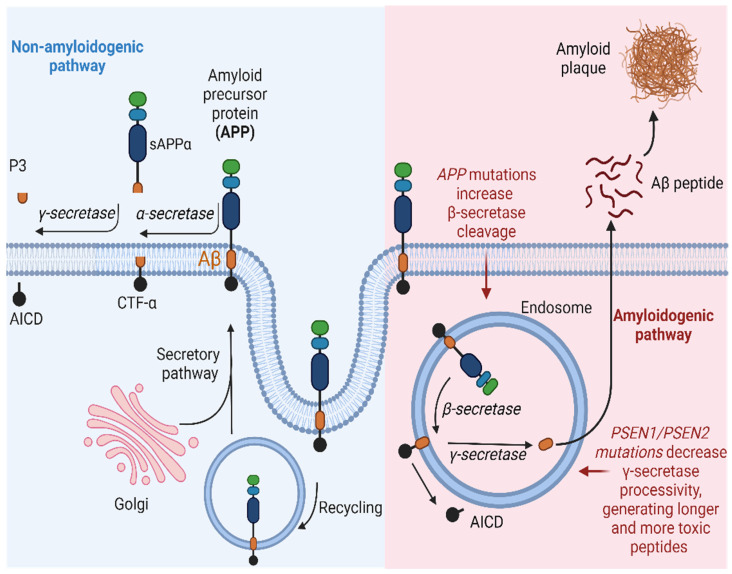
Schematic representation of the amyloidogenic and non-amyloidogenic pathways. (**Left**) APP undergoes proteolytic cleavage by α-secretase, generating sAPPα and CTFα. Subsequent processing by γ-secretase yields p3 and AICD. This non-amyloidogenic pathway occurs primarily at the plasma membrane, with APP recycling in endosomes facilitated by interaction with low-density lipoprotein receptors. (**Right**) In the amyloidogenic pathway, APP is cleaved by β-secretase, releasing sAPPβ and CTFβ. Further cleavage by γ-secretase results in the release of Aβ and AICD. This pathway predominantly occurs within endosomes. AICD translocates to the nucleus, functioning as a transcriptional factor, while extracellular Aβ monomers aggregate, forming oligomers or fibrils/plaques associated with AD. Created with BioRender.com.

**Figure 4 diseases-12-00110-f004:**
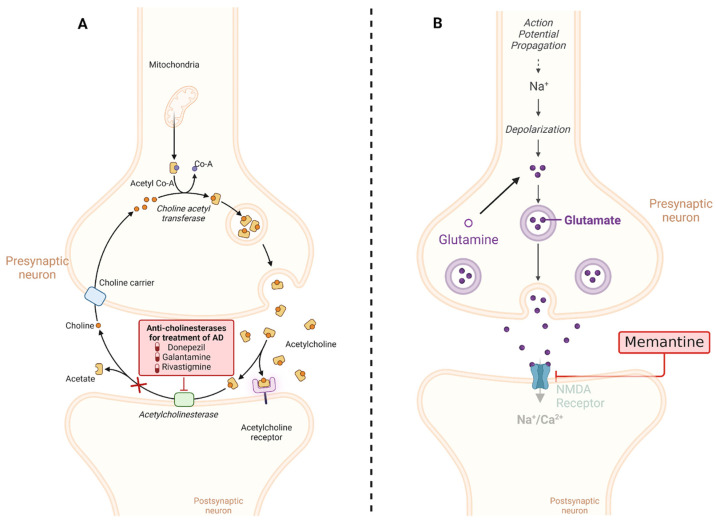
Mechanisms of action of the most common drugs used in AD. (**A**) CIs such as donepezil, rivastigmine, and galantamine work by blocking the activity of acetylcholinesterase, an enzyme that breaks down acetylcholine. By inhibiting acetylcholinesterase, these medications increase the levels of acetylcholine in the brain, which can help improve cognitive function in AD patients. (**B**) Memantine is an NMDA receptor antagonist that regulates glutamate activity in the brain. By blocking excessive activation of NMDA receptors, memantine helps to protect neurons from excitotoxicity, which is associated with neurodegenerative processes in AD. This mechanism helps to stabilize neuronal activity and may slow the progression of cognitive decline in AD patients. This diagram illustrates the distinct mechanisms of action of cholinesterase inhibitors and memantine in the treatment of AD. Created with BioRender.com.

**Figure 5 diseases-12-00110-f005:**
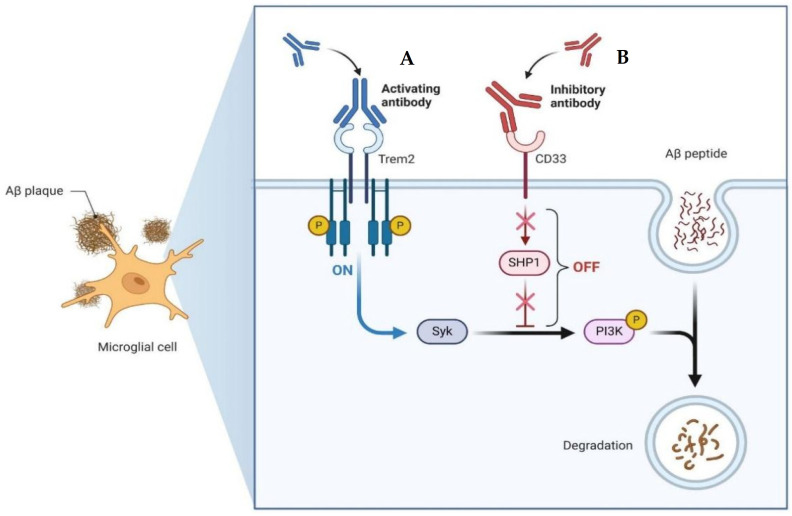
Example of the mechanisms of action of an emerging treatment with antibodies for AD. (**A**) Activating antibody for TREM2. This antibody targets TREM2 and interacts with Syk. By activating TREM2 signaling pathways via Syk, this antibody enhances microglial function and promotes the clearance of Aβ plaques from the brain. (**B**) Inhibitory antibody for CD33. By interacting with SHP1, the antibody inhibits the inhibitory signaling pathways downstream of CD33. This results in enhanced microglial phagocytic activity and degradation of Aβ. Both antibodies offer promising therapeutic approaches for AD by targeting key immune regulatory proteins involved in the clearance of Aβ plaques. Created with BioRender.com.

**Figure 6 diseases-12-00110-f006:**
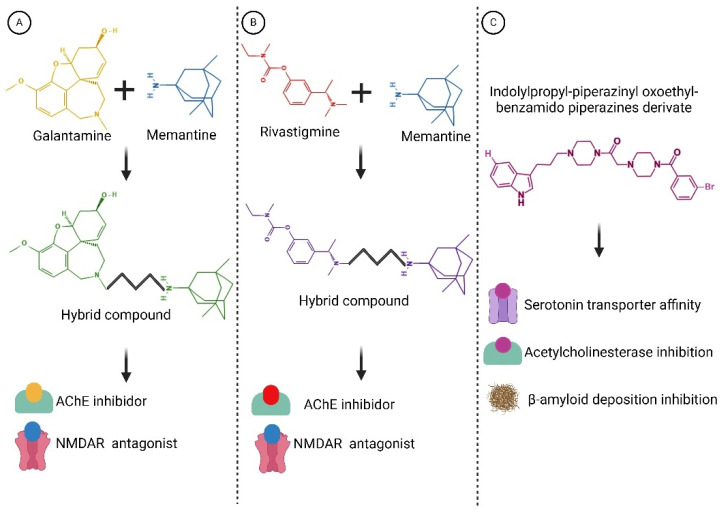
Structures of hybrid compounds and indolylpropyl benzamidopiperazine derivate. (**A**) Galantamine and memantine. (**B**) Rivastigmine and memantine. (**C**) indolylpropyl benzamidopiperazine derivate. The hybrid compounds are shown by the color combination of the drugs. Functions are indicated below. These compounds represent a promising class of multitarget drugs, potentially leading to a reduction in its progression. Created with BioRender.com.

**Figure 7 diseases-12-00110-f007:**
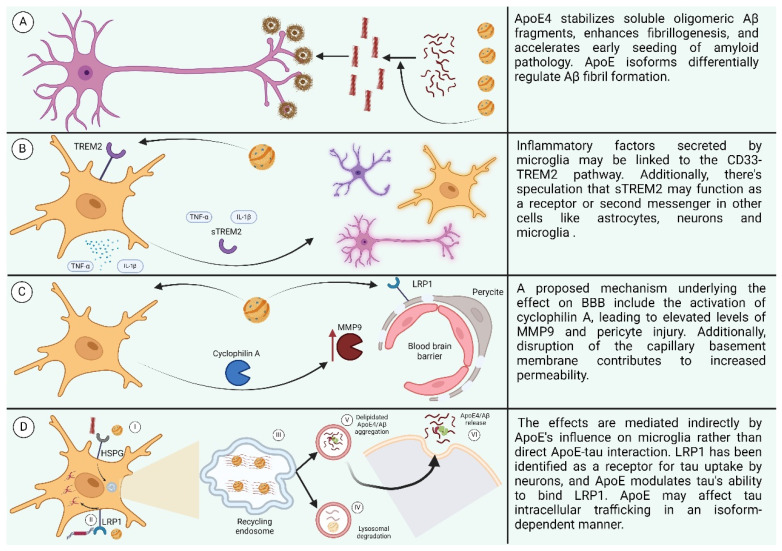
(**A**) Effect of ApoE4 on AB. (**B**) Effect of ApoE4 on neuroinflammation. (**C**) Effect of ApoE4 on BBB. (**D**) Effect of ApoE on receptors. (I: ApoE4 and Aβ oligomers compete for binding to HSPGs. II: LRP1 interacts with lipidated ApoE4, leading to its internalization. III: Recycling endosomes serve as sites for Aβ generation. IV: Aβ and lipidated ApoE4 undergo degradation in lysosomes. V: ApoE4 stimulates the aggregation of Aβ. VI: Aggregated ApoE4 and Aβ are released into the extracellular space.) Created with BioRender.com.

**Table 1 diseases-12-00110-t001:** Mainly differences between EOAD and LOAD.

Characteristics	EOAD	LOAD
Age at onset	<65 years	≥65 years
Sex distribution	Female = male	Female > male
Atypical signs and symptoms	Higher prevalence of dysexecutive symptoms and atypical presentations	Uncommon
Episodic memory loss	Early (late in atypical cases)	Early
Genetic contributions	Probably polygenic and autosomal dominant in some cases	Probably polygenic
Heritability	92–100%	60–80%
Neuropathological hallmarks	Plaques and tangles	Plaques and tangles
Initial PET signal evidence	Precuneus and mesial temporal region	Mesial temporal region
Initial Aβ accumulation	Neocortex (striatal in some autosomal dominant cases)	Neocortex
Rate of brain atrophy	Fast	Low
Tau burden	Higher in cortical and stratum areas	Higher in limbic areas
Effect of APOE genotype	*APOEε4*: Accelerates/diminishes cognitive decline depending on other factors [140,167,168,169,175,176,177]. *APOEε2*: Significantly delays the age of onset [178]. *APOEε3*: Further research is needed, although a rare variant has been described to exert a protective effect [133].	*APOEε4*: Accelerates cognitive decline and especially promotes the appearance of amnesic phenotypes [129,130,131,132,134,135,136,137]. *APOEε2*: Slows general cognitive rate decline [132,142]. *APOEε3*: Is considered a neutral risk factor for AD development.
Pharmacological therapy	Therapies focus on the underlying pathological mechanisms of the disease. Patients may benefit from gene-based therapies aimed at reducing the production or aggregation of Aβ [179]	Therapies primarily focus on managing symptoms and improving cognitive function rather than targeting the underlying disease process as well as Gamma-Secretase modulators. In these patients, a gene-based therapy may not be very useful [100,102,103,104,105,106,107,108,109,110]

Modified from Korczyn and Grinberg, 2024 [180].

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
