# Peer review of "Early- and Late-Onset Alzheimer’s Disease: Two Sides of the Same Coin?"

_diseases, 2024, doi:10.3390/diseases12060110_

Round 1
Reviewer 1 Report
Comments and Suggestions for Authors
It is an interesting idea to write a review on Alzheimer's disease, focusing on side-by-side comparison between the early-onset and late-onset forms of the disease. I would like to suggest revising sections 5 (Treatments and Therapeutic Approaches) and 6 (The Effect of APOE).
Section 5: Please do not put Memantine under "5.1. Cholinesterase Inhibitors" but rather create a new section (5.2. NMDA Receptor Inhibitor) to describe Memantine.
Section 5.1. , lines 374-375. You should indicate that the lack of ACh is also due to the selective death of cholinergic neurons.
Memantine paragraph, lines 398-399. You should indicate that the NMDA receptor is chronically activated by Abeta oligomers.
Section 5.2. (Emerging Treatments). You should have the first paragraph talk about the anti-Abeta antibody drugs (already approved) before talking about drugs that are still in the pipeline.
Same section lines 422-423 "...the goal is to enhance mitochondrial activity..." Here, you should probably add a discussion about how activation of microglia is both good (Abeta clearance) and bad (neuroinflammation).
Section 6. The first 2 paragraphs (lines 431-441) are out of place. I suggest starting with the next paragraph (lines 443-453). I would also suggest adding a stronger justification for focusing on APOE and the differential impact it has on EOAD and LOAD.
Same section, since you are devoting a whole section on APOE, I suggest including clear, concise background on APOE, such as what cells produce it, where is it located, what is its physiological function, what is the structure of the different isoforms, where are the binding domains, etc.
Section 7. (Future Perspectives), lines 640-643. The point of using APOE4 as drug target is well taken but I would also be interested to know if mimicking the effect of protective APOE2 is also being explored?
Comments on the Quality of English LanguageEnglish writing is fine, very few mistakes/typos detected.
Author Response
Dear Reviewer,
Thank you very much for your thorough review and insightful suggestions regarding our manuscript entitled "Early- and late-onset Alzheimer's disease: two sides of the same coin?". We greatly appreciate your time and attention to detail.
Based on your feedback, we have made several revisions to the manuscript, which we will detail below:
Section 5: Please do not put Memantine under "5.1. Cholinesterase Inhibitors" but rather create a new section (5.2. NMDA Receptor Inhibitor) to describe Memantine.
We have created a new subsection, "5.1.2 NMDA receptor inhibitor," detailing the mechanism of action of Memantine. Additionally, we have divided Section 5 into conventional treatments and emerging treatments, as suggested.
Section 5.1. , lines 374-375. You should indicate that the lack of ACh is also due to the selective death of cholinergic neurons.
The role of selective death of cholinergic neurons in the lack of ACh has been added in lines 491-495.
Memantine paragraph, lines 398-399. You should indicate that the NMDA receptor is chronically activated by Abeta oligomers.
We have included information about the chronic activation of the NMDA receptor by Abeta oligomers in lines 525-529.
Section 5.2. (Emerging Treatments). You should have the first paragraph talk about the anti-Abeta antibody drugs (already approved) before talking about drugs that are still in the pipeline.
We have added a paragraph discussing anti-Abeta antibody drugs, including those already approved, before discussing drugs in the pipeline. Furthermore, we have included information on additional drugs undergoing clinical trials.
Same section lines 422-423 "...the goal is to enhance mitochondrial activity..." Here, you should probably add a discussion about how activation of microglia is both good (Abeta clearance) and bad (neuroinflammation).
A discussion on the dual role of microglia activation in Abeta clearance and neuroinflammation has been added in lines 579-586.
Section 6. The first 2 paragraphs (lines 431-441) are out of place. I suggest starting with the next paragraph (lines 443-453). I would also suggest adding a stronger justification for focusing on APOE and the differential impact it has on EOAD and LOAD. Same section, since you are devoting a whole section on APOE, I suggest including clear, concise background on APOE, such as what cells produce it, where is it located, what is its physiological function, what is the structure of the different isoforms, where are the binding domains, etc.
We have restructured Section 6, starting with an introduction to ApoE and detailing the molecular differences of its isoforms in lines 636-651. Additionally, we have provided a stronger justification for focusing on ApoE and its impact on both EOAD and LOAD.
Section 7. (Future Perspectives), lines 640-643. The point of using APOE4 as drug target is well taken but I would also be interested to know if mimicking the effect of protective APOE2 is also being explored?
We have included your excellent suggestion about exploring the mimicking effect of protective ApoE2 in addition to targeting ApoE4 as a drug target in lines 846-852.
Once again, we truly appreciate your valuable feedback, which has undoubtedly improved our manuscript. Please let us know if there are any further revisions or clarifications needed.
Reviewer 2 Report
Comments and Suggestions for Authors
In their review, Valdez-Gaxiola give an overview on the current clinical progress in Alzheimer’s disease, comparing early onset with late onset pathology. Although this is a very elaborate review, the focus is on a more general information related to both molecular, genetic and clinical aspects of AD; hence not really for basic scientists, but a rather broader public of clinicians (I guess).
As such, certain aspects seem not too accurate or reflecting the state of the art in literature and if the authors aim for a timely review, they should include these updates in a revised version.
- With respect to causal genes, besides APP, PSEN1 and PSEN2, SorlA appears to get as well the status of a causal gene and should be added (literature of Small, Petsko, Holstege labs). Also certain genes are coming close to that status as well and which are relatively rare in the population with confer a relatively significant risk: examples are TREM2, ABCA7, ADAM10. For an updated review please consult Scheltens et al, The Lancet 2021.
- With respect to LOAD, the authors limit the discussion to mentioning a few risk loci (CLU, CR1, PICALM, BIN1 etc), but the genetic landscape is far more evolved (again, see Scheltens et al; but also Van Acker et al., Mol. Neurodeg, 2019).
- With respect on how FAD mutations in APP and PSENs impact on AD, the info is outdated. It has been demonstrated by now that mutations in PSEN1 do not increase activity, but decrease processivity, generating longer peptides. Except for for instance the Swedish mutation in APP, it is mostly not about producing more Abeta but shifting the production to longer, more toxic forms (in many cases even resulting in less Abeta). Info on figure 3 is outdated/not complete in this respect. See literature of the Chavez-Gutierrez lab (Szaruga et al., Cell, 2017). Also, there is no mentioning about the recent couple of papers (from the Chavez-Gutierrez and Selkoe labs; eg. Petit et al Mol Psych, 2022; Liu et al. Alz Dem 2023) demonstrating a very strong correlation between age of onset and decreased processivity.
- With respect to organellar dysfunction, the authors only mention mitochondrial dysfunction and its relevance for synaptic function. However, a major hallmark at early, preclinical stages, is endolysosomal defects that may trigger aberrant APP processing and clearance (in neurons and microglia resp; see eg Peric & Annaert, Acta Neuropath., 2015; Van Acker et al. Ageing Research reviews 2022). Herein, this is more and more becoming linked to the toxic effects of accumulating APP-CTFs (as the substrate of g-secretase), and more recently including the underlying mechanism of APP-CTF toxicity: see for instance Im et al. Sci Adv, 2023; Bretou et al., DevCell 2024; Kwart et al Neuron 2019 ; Hung et al Cell Rep 2021 ; Hung and Livesey, Autophagy 2021).
- With respect to curing AD, the emphasis is again on ‘old’ drugs, like cholinesterase inhibitors and NMDA antagonists. The authors completely miss the major progress we are making in clearing Abeta and the very promising clinical trials on Lecanemab, docanemab etc. In this respect, there is as well an increasing awareness and attention to g-secretase modulators as these will stabilize g-secretase processing towards shorter, harmless peptides.
Author Response
Dear Reviewer,
Thank you for your thorough review and insightful comments on our manuscript. We appreciate the time you've taken to provide detailed feedback, which has been invaluable in enhancing the quality and comprehensiveness of our work.
In response to your suggestions, we have made the following revisions:
With respect to causal genes, besides APP, PSEN1 and PSEN2, SorlA appears to get as well the status of a causal gene and should be added (literature of Small, Petsko, Holstege labs). Also certain genes are coming close to that status as well and which are relatively rare in the population with confer a relatively significant risk: examples are TREM2, ABCA7, ADAM10. For an updated review please consult Scheltens et al, The Lancet 2021.
With respect to LOAD, the authors limit the discussion to mentioning a few risk loci (CLU, CR1, PICALM, BIN1 etc), but the genetic landscape is far more evolved (again, see Scheltens et al; but also Van Acker et al., Mol. Neurodeg, 2019)
We have expanded the discussion to include references to studies evaluating the polygenic risk in both Early-Onset Alzheimer's Disease (EOAD) and Late-Onset Alzheimer's Disease (LOAD) (lines 101-122). Additionally, we have highlighted the importance of acknowledging more genes involved in the pathogenesis of AD.
With respect on how FAD mutations in APP and PSENs impact on AD, the info is outdated. It has been demonstrated by now that mutations in PSEN1 do not increase activity, but decrease processivity, generating longer peptides. Except for for instance the Swedish mutation in APP, it is mostly not about producing more Abeta but shifting the production to longer, more toxic forms (in many cases even resulting in less Abeta). Info on figure 3 is outdated/not complete in this respect. See literature of the Chavez-Gutierrez lab (Szaruga et al., Cell, 2017). Also, there is no mentioning about the recent couple of papers (from the Chavez-Gutierrez and Selkoe labs; eg. Petit et al Mol Psych, 2022; Liu et al. Alz Dem 2023) demonstrating a very strong correlation between age of onset and decreased processivity.
We have further developed the discussion on the impact of Familial Alzheimer's Disease (FAD) mutations in APP and PSENs, incorporating updated information from the literature (lines 299-303). We have also revised Figure 3 to ensure its accuracy and completeness in this regard.
With respect to organellar dysfunction, the authors only mention mitochondrial dysfunction and its relevance for synaptic function. However, a major hallmark at early, preclinical stages, is endolysosomal defects that may trigger aberrant APP processing and clearance (in neurons and microglia resp; see eg Peric & Annaert, Acta Neuropath., 2015; Van Acker et al. Ageing Research reviews 2022). Herein, this is more and more becoming linked to the toxic effects of accumulating APP-CTFs (as the substrate of g-secretase), and more recently including the underlying mechanism of APP-CTF toxicity: see for instance Im et al. Sci Adv, 2023; Bretou et al., DevCell 2024; Kwart et al Neuron 2019 ; Hung et al Cell Rep 2021 ; Hung and Livesey, Autophagy 2021).
Organellar dysfunction, particularly endolysosomal defects and their implications for aberrant APP processing and clearance, has been addressed in more detail, with references to relevant studies (lines 213-240).
With respect to curing AD, the emphasis is again on ‘old’ drugs, like cholinesterase inhibitors and NMDA antagonists. The authors completely miss the major progress we are making in clearing Abeta and the very promising clinical trials on Lecanemab, docanemab etc. In this respect, there is as well an increasing awareness and attention to g-secretase modulators as these will stabilize g-secretase processing towards shorter, harmless peptides.
We have expanded the discussion to include emerging treatments such as monoclonal antibodies and gamma-secretase modulators, highlighting the promising clinical trials on drugs like Lecanemab and docanemab (newly included in section 5.2 Emerging Treatments) lines 540-566 and 617-633.
These revisions aim to provide a more comprehensive and up-to-date overview of the genetic, cellular, and therapeutic aspects of Alzheimer's disease. We believe that these enhancements significantly improved the manuscript and better serve the needs of our readership.
Once again, we sincerely appreciate your valuable feedback, which has greatly contributed to the improvement of our work.
Reviewer 3 Report
Comments and Suggestions for Authors
The authors of « Early- and late-onset Alzheimer's disease: two sides of the same coin? » have done an extensive review of the main differences on the early and late on appearance on life of Alzheimer’s disease, with a brief historical explanation and the difficulties to study and compare them in a multifactorial and complex disease as it is AD.
In general, I like their writing style and explanations are clear enough to be understood. Nevertheless, I would arise a few points that I think it could discussed in benefit of the paper.
On line 111-112, it seems like a sub header of the section since it is introducing the non-genetic risk factors, should it be bold? A subsection 2.1?
On line 125, “Smoking: is a modifiable risk factor for LOAD”, I guess that smoking should be bold, but also the use of the adjective “modifiable” makes the phrase harder to understand. Can do they find an alternative? In addition, I am not sure that it is more modifiable than diet (as example) or other risk factors, or do they mean something else that I do not understand?
Figure legends of figure 2 & 3 are too short and does not described fully the figure. Figure 3, the non-amyloidogenic pathway is not a pathological mechanism. It is the imbalance or shift between both pathways. I think they should be explained properly. In addition, Figure 4 legend is not complete on my file, and probably neither Fig. 6.
On line 620, the authors claim, “for EOAD, which tends to receive less attention in research and public discourse compared to LOAD”. I will not discuss about the public discourse, but I think that’s not the case on basic research where EOAD probably is over-represented due to that most of the animal models are based on the genetic factors. Maybe do they mean clinical research?
Finally, on line 676, data availability, I do not think that it is applicable on a review.
Author Response
Dear reviewer,
Thank you for your insightful comments on our manuscript. We have carefully addressed each of your suggestions and made the necessary revisions to improve the clarity and comprehensibility of the content.
On line 111-112, it seems like a sub header of the section since it is introducing the non-genetic risk factors, should it be bold? A subsection 2.1?
We agree that it would improve clarity to treat the introduction of non-genetic risk factors as a subsection. Therefore, we have divided the section into three subsections: 2.1 genetic factors, 2.2 non-genetic factors, and 2.3 other factors, as you suggested.
On line 125, “Smoking: is a modifiable risk factor for LOAD”, I guess that smoking should be bold, but also the use of the adjective “modifiable” makes the phrase harder to understand. Can do they find an alternative? In addition, I am not sure that it is more modifiable than diet (as example) or other risk factors, or do they mean something else that I do not understand?
We have revised it to improve clarity and removed the use of the term "modifiable" to avoid ambiguity, L 160-162.
Figure legends of figure 2 & 3 are too short and does not described fully the figure. Figure 3, the non-amyloidogenic pathway is not a pathological mechanism. It is the imbalance or shift between both pathways. I think they should be explained properly. In addition, Figure 4 legend is not complete on my file, and probably neither Fig. 6.
We have added detailed descriptions to Figures 2, 3, 4, and 5 to provide a better understanding of the content they represent, also we added Figure 6.
On line 620, the authors claim, “for EOAD, which tends to receive less attention in research and public discourse compared to LOAD”. I will not discuss about the public discourse, but I think that’s not the case on basic research where EOAD probably is over-represented due to that most of the animal models are based on the genetic factors. Maybe do they mean clinical research?
We have rephrased the statement to emphasize the public domine, particularly in lines 818-824, while acknowledging that EOAD may indeed receive attention in basic research due to the prevalence of genetic factors.
Finally, on line 676, data availability, I do not think that it is applicable on a review.
We have removed the discussion on data availability, as it is not applicable to a review.
We believe that these revisions have improved the clarity and comprehensiveness of our manuscript. Thank you once again for your valuable feedback, which has undoubtedly improved the quality of our work.
Reviewer 4 Report
Comments and Suggestions for Authors
The authors reviewed the literature about Alzheimer's disease with the aim at dissecting disease's features, mechanisms, and treatments of early and late onsets.
The aim of the review is worth, but the definition of the difference remain elusive.
The reader does not understand what is really new in the manuscript.
For instance most of the important data have been already reported in Ref. 147.
The main conclusion about mechanistic difference is Section 3.3:
but this is basically expected as the difference between familial and sporadic AD.
It is already known that LOAD is more sporadic than EOAD, the latter more genetic.
The opinion of this reviewer is that the review does not add relevant information nor a further organization of the existing one.
Therefore, I recommend the rejection of the manuscript.
Some more details are below.
L.110 - The so-called scaffolding effect should be introduced here.
See for instance:
https://doi.org/10.1146/annurev.psych.59.103006.093656
Scaffolding is expected to be conditionally more effective in LOAD than in EOAD.
This explains at least in part the heterogeneous character of LOAD.
L.117 - ", due ..."
L.141 - Introduce miRNA and lncRNA.
Fig.1 should be moved up, belonging to section 2.
L.163 - Neurofibrillary tangles contain hyperphosphorylated tau protein.
Chemical modifications of tau are important early events in neurodegeneration.
L.171 - APOE alleles are usually indicated with ApoE-epsilonX, with
X=2,3,4.
See L.209.
L.177-178 - It is not clear what "internalization, and aggregation" refer to.
The hypothesis of tau protein diffusion is not confirmed yet and some discussion is deserved here.
See ref.38.
The authors correctly describe the prion-like propagation as a proposal (see L.224-244).
L.214 - See comment above (L.163).
Section 3.3 -
The "purer" hallmarks of AD (plaques and tangles) in EOAD can be directly linked to genetic causes (Figs.1 and 3: APP, PSEN1, PSEN2 variants).
Then EOAD is clearer than LOAD since the beginning of the manuscript.
Section 4 -
There are problems in sub-section headings.
Figure 4 - Caption is incomplete.
Same for Fig.5.
Section 4.5 -
Biomarkers are potentially good targets to discrimanate among EOAD and LOAD.
However, the section does not add any useful information to accomplish the task.
Sections 5-6 -
The sections describe pathways that are likely affecting LOAD.
When EOAD is concerned (for example APOE effects) everything becomes uncertain.
L. 620-621 - It is not clear why EOAD, occurring in about 5% of AD cases (Fig.1), "...can be equally devastating" when compared to LOAD.
This sentence should be supported by an observed increasing EOAD occurrence.
The main issue is the age of onset (AoO), where 65 years is a convention based on the actual life expectancy.
If life expectancy increases, AoO should adapt to it.
Conclusions -
The whole manuscript converges on the homogeneity of EOAD compared to heterogeneity of LOAD.
The readers reaches the end of the manuscript convinced that EOAD is not elusive like LOAD.
This conclusion is shared among the existing literature.
Then the questions in L.659-666 are marginal compared to the surmounting problems of understanding how to cope with LOAD.
Only minor corrections required.
Author Response
Dear Reviewer,
Thank you for your detailed review of our manuscript entitled "Early- and late-onset Alzheimer's disease: two sides of the same coin?". We sincerely appreciate the time and effort you've dedicated to providing valuable feedback.
We understand your concerns about the novelty of our review and the clarity of the differences between early- and late-onset Alzheimer's disease (EOAD and LOAD). Allow us to address your points and hopefully convince you of the importance of the topic:
Highlighting Important Findings: While it's true that much of the information presented in our review has been reported previously, our goal was to synthesize and highlight the most significant findings in the field over time, with particular emphasis on the comparison between EOAD and LOAD. In doing so, we aim to provide the readers with a comprehensive understanding of the disease’s characteristics, mechanisms, and treatment options.
Mechanistic Differences: We acknowledge that the mechanistic differences between familial and sporadic AD are well established. However, our review delves deeper into these differences, focusing specifically on the distinct characteristics of EOAD and LOAD. Despite LOAD being predominantly sporadic and EOAD having a stronger monogenic component, our analysis reveals nuances that contribute to the heterogeneous nature of the disease.
Addressing Specific Points: We have carefully addressed each of your detailed comments and suggestions for improvement.
L.110 - The so-called scaffolding effect should be introduced here.
See for instance: https://doi.org/10.1146/annurev.psych.59.103006.093656
Scaffolding is expected to be conditionally more effective in LOAD than in EOAD.
This explains at least in part the heterogeneous character of LOAD.
We have added scaffolding effect to improve the comprehensiveness of our manuscript (L 135-147).
L.117 - ", due ..."
Corrected
L.141 - Introduce miRNA and lncRNA.
We have expanded the discussion on miRNAs and lncRNAs in lines 171-205.
Fig.1 should be moved up, belonging to section 2.
The figure has been relocated accordingly.
L.163 - Neurofibrillary tangles contain hyperphosphorylated tau protein.
We have included this information in lines 253-257.
L.171 - APOE alleles are usually indicated with ApoE-epsilonX, with
X=2,3,4.
See L.209.
We have clarified the notation for APOE alleles we have changed APOEεX for the genotype and ApoEX for the protein, since the Human Genome Organization website mentions the nomenclature of the APOE allele and that most articles on the subject use ApoE to name the protein
L.177-178 - It is not clear what "internalization, and aggregation" refer to.
We have provided further clarification in lines 337-343 and acknowledged that the hypothesis of tau protein diffusion is still under discussion (line 321). Additionally, we referenced the suggested literature (ref.38).
Section 4 -
There are problems in sub-section headings.
Sub-section headings have been corrected.
Figure 4 - Caption is incomplete.
Same for Fig.5.
Detailed captions have been added to all figures.
Section 4.5 - Biomarkers are potentially good targets to discrimanate among EOAD and LOAD.
However, the section does not add any useful information to accomplish the task.
We have expanded the discussion on biomarkers in lines 450-461 to provide more useful information for discriminating between EOAD and LOAD.
Sections 5-6 - The sections describe pathways that are likely affecting LOAD.
When EOAD is concerned (for example APOE effects) everything becomes uncertain.
Indeed, the understanding of how APOE impacts EOAD is relatively limited compared to its effects on LOAD. This lack of clarity underscores the importance of further investigation into this area. By discussing APOE in the context of both EOAD and LOAD, we aim to emphasize to readers the gap in knowledge and the need for continued research in this field.
- 620-621 - It is not clear why EOAD, occurring in about 5% of AD cases (Fig.1), "...can be equally devastating" when compared to LOAD.
We have provided justification for this statement in line 824.
Conclusions: While it's true that our manuscript converges on the homogeneity of EOAD compared to the heterogeneity of LOAD, we believe this conclusion is significant as it highlights the distinct challenges associated with each form of the disease. By elucidating these differences, we contribute to the broader understanding of Alzheimer's disease, which is crucial for the design and development of effective therapeutic strategies.
We believe that our manuscript provides valuable insights into the complexities of Alzheimer's disease, particularly in elucidating the differences between EOAD and LOAD. We hope that with the revisions and clarifications we've made, you will reconsider your recommendation and recognize the contribution our work makes to the scientific literature.
Thank you once again for your thoughtful review.
Reviewer 5 Report
Comments and Suggestions for Authors
Major comments:
It would be desirable for the authors to include a figure depicting the chemical structures of some notable new compounds designed for Alzheimer's disease (A.D.) treatment. For example, hybrid molecules with dual affinity for acetylcholinesterase (AchE) and serotonin transporter (SERT) could be illustrated (for reference, see: https://doi.org/10.1016/j.ejmech.2020.112368). This information could be incorporated into Part 5 of the review, specifically under "Treatments and Therapeutic Approaches."
In general, the figures exhibit excellent quality, but there is a misconfiguration in Figure 1, where the legend is placed incorrectly. Please rectify this (Page 4, Figure 1).
The same issue is observed in Figure 2 (Page 7, lines 220-221; the legend is misplaced).
In a paper, figures should be self-explanatory. Readers should not need to refer to the main text to understand them. Therefore, it is recommended to add more detailed legends for Figures 2, 3, 4, and 5.
Figure 5 is misconfigured, with the legend positioned atop the image.
In Table 1, the authors summarize the main differences between early-onset Alzheimer's disease (EOAD) and late-onset Alzheimer's disease (LOAD). It is suggested to include a row detailing "Pharmacological therapy" for both cases.
In the "Future Perspectives" section, perhaps a comment on the potential preventive role of dietary anti-Alzheimer's disease interventions could be added (e.g., https://doi.org/10.1001/archneurol.2010.84 ; https://doi.org/10.3389/fnut.2021.688086).
Author Response
Dear Reviewer,
Thank you for your insightful comments on our manuscript. We have carefully addressed each of your suggestions and made the necessary revisions to improve the clarity and comprehensibility of the content.
It would be desirable for the authors to include a figure depicting the chemical structures of some notable new compounds designed for Alzheimer's disease (A.D.) treatment. For example, hybrid molecules with dual affinity for acetylcholinesterase (AchE) and serotonin transporter (SERT) could be illustrated (for reference, see: https://doi.org/10.1016/j.ejmech.2020.112368). This information could be incorporated into Part 5 of the review, specifically under "Treatments and Therapeutic Approaches."
We have included Figure 6 depicting hybrid molecules with dual affinity for Alzheimer's disease treatment, as you suggested.
In general, the figures exhibit excellent quality, but there is a misconfiguration in Figure 1, where the legend is placed incorrectly. Please rectify this (Page 4, Figure 1).
The same issue is observed in Figure 2 (Page 7, lines 220-221; the legend is misplaced).
Figure 5 is misconfigured, with the legend positioned atop the image.
We have rectified the misconfigurations in Figures 1, 2, and 5, ensuring that the legends are correctly positioned.
In a paper, figures should be self-explanatory. Readers should not need to refer to the main text to understand them. Therefore, it is recommended to add more detailed legends for Figures 2, 3, 4, and 5.
To enhance the self-explanatory nature of the figures, we have added more detailed legends to Figures 2, 3, 4, and 5, as recommended.
In Table 1, the authors summarize the main differences between early-onset Alzheimer's disease (EOAD) and late-onset Alzheimer's disease (LOAD). It is suggested to include a row detailing "Pharmacological therapy" for both cases.
We have included a row detailing pharmacological therapy for both EOAD and LOAD, per your suggestion.
In the "Future Perspectives" section, perhaps a comment on the potential preventive role of dietary anti-Alzheimer's disease interventions could be added (e.g., https://doi.org/10.1001/archneurol.2010.84 ; https://doi.org/10.3389/fnut.2021.688086).
We have incorporated a comment on the potential preventive role of dietary anti-Alzheimer's disease interventions, as you proposed (L 852-859).
We believe that these revisions have improved the manuscript and improved its overall quality. Thank you once again for your valuable feedback and guidance throughout this process.
Round 2
Reviewer 4 Report
Comments and Suggestions for Authors
The authors added to the manuscript many references and several discussions aiming at clarifying the specificity of early-onset Alzheimer's disease compared to late one.
The main criticism I had on this review, about new information provided, still persists.
Once EOAD is addressed to genetic causes, the single novelty is gene editing by the recent CRISPR technique (added lines 848-851), but no references are provided. The effect of small molecules (ref. 179) is a reasonable hypothesis, not confirmed by successfull therapies, yet. The certain effect of physical activity and life-style changes are demonstrated basically for LOAD only, acting over about 10-20 years (ref. 29, for instance). I do not find any reference where physical activity is found effective on EOAD.
The following conclusion (lines 862-864) about the need of shifting upwards the age of onset is related to LOAD prevention. The required shift is the reason of "devastating" character of AD in recent years. There are no points in the manuscript where an indication of increase in EOAD is put in evidence. It is true (line 821) that the most solid information about AD is its genetic cause, but it is also true that the "disproportionate" of LOAD is justified by the shifting of AoO.
Therefore, I am still convinced that the review does not add relevant information to the field.
Comments on the Quality of English LanguageOnly minor corrections that can be cleared in proof are in the revised manuscript.